# Cross-regulation and cross-talk of conserved and accessory two-component regulatory systems orchestrate *Pseudomonas* copper resistance

**Sylvie Elsen** ®*, **Victor Simon** ®¤, **Ina Attrée**®

University Grenoble Alpes, Institute of Structural Biology, UMR5075, Team Bacterial Pathogenesis and Cellular Responses, Grenoble, France

¤ Current address: Université de Lyon, INSA Lyon, Université Claude Bernard Lyon 1, CNRS UMR5240, Laboratoire de Microbiologie, Adaptation et Pathogénie, Villeurbanne, France
* sylvie.elsen@ibs.fr

**Data Availability Statement:** All data are in the manuscript and/or supporting information files.

**Funding:** This work was supported by Agence Nationale de la Recherche: ANR-17-EURE-0003

## Abstract

Bacteria use diverse strategies and molecular machinery to maintain copper homeostasis and to cope with its toxic effects. Some genetic elements providing copper resistance are acquired by horizontal gene transfer; however, little is known about how they are controlled and integrated into the central regulatory network. Here, we studied two copper-responsive systems in a clinical isolate of *Pseudomonas paraeruginosa* and deciphered the regulatory and cross-regulation mechanisms. To do so, we combined mutagenesis, transcriptional fusion analyses and copper sensitivity phenotypes. Our results showed that the accessory CusRS two-component system (TCS) responds to copper and activates both its own expression and that of the adjacent nine-gene operon (the *pcoA2* operon) to provide resistance to elevated levels of extracellular copper. The same locus was also found to be regulated by two core-genome-encoded TCSs—the copper-responsive CopRS and the zinc-responsive CzcRS. Although the target palindromic sequence–ATTCATnnATGTAAT–is the same for the three response regulators, transcriptional outcomes differ. Thus, depending on the operon/regulator pair, binding can result in different activation levels (from none to high), with the systems demonstrating considerable plasticity. Unexpectedly, although the classical CusRS and the noncanonical CopRS TCSs rely on distinct signaling mechanisms (kinase-based *vs.* phosphatase-based), we discovered cross-talk in the absence of the cognate sensory kinases. This cross-talk occurred between the proteins of these two otherwise independent systems. The *cusRS-pcoA2* locus is part of an Integrative and Conjugative Element and was found in other *Pseudomonas* strains where its expression could provide copper resistance under appropriate conditions. The results presented here illustrate how acquired genetic elements can become part of endogenous regulatory networks, providing a physiological advantage. They also highlight the potential for broader effects of accessory regulatory proteins through interference with core regulatory proteins.

through the Laboratory of Excellence GRAL and the University Grenoble, Alpes graduate school (Ecoles Universitaires de Recherche) CBH-EUR-GS. We further acknowledge support from CNRS, CEA and Grenoble Alpes University. Ph.D. fellowship for VS was funded by the French Ministry of Education and Research. The funders had no role in study design, data collection and analysis, decision to publish, or preparation of the manuscript.

**Competing interests:** The authors have declared that no competing interests exist.

## Author summary

Two-component regulatory systems play a key role in bacterial life by detecting and integrating a wide range of signals, allowing bacteria to continuously monitor and adapt to a changing environment. They weave a complex network with a few highly interconnected phosphorelays and numerous cross-regulations. Our study reveals connections between zinc and copper homeostasis in a pathogenic bacterium, with cross-regulation observed for three independent and closely related transcriptional regulators, CzcR, CopR and CusR. Zinc and copper play a major role in host-pathogen interactions, and bacteria that synergize their responses to the two elements can harness a growth advantage and enhanced fitness in specific conditions. We also observed unexpected cross-talk between the core genome-encoded CopRS system and the horizontally acquired CusRS system, although the phosphorylation state of the response regulators is controlled differently by cognate sensory kinases. This plasticity observed within the two signaling systems ensures a normal regulatory response to the copper signal required to maintain copper homeostasis.

## Introduction

Copper can be considered a "Dr. Jekyll and Mr. Hyde" heavy metal, as it is an essential trace element in biological systems but, in excess, generates free radicals with deleterious consequences [1,2]. As an antimicrobial, copper is extensively used to treat crops, to enhance breeding, and as part of medicine, with the first reports of use dating back to ancient civilizations. Human physiology exploits its properties, notably by accumulating copper in phagocytic cells and at infection sites where it contributes to protection against viruses, fungi and bacteria [3]. However, bacteria also use copper and have developed mechanisms to deal with its excess.

To maintain copper homeostasis, Gram-negative bacteria use various molecular mechanisms and machinery [1,2,4,5]. Cuproteins are essentially extracellular or periplasmic, participating in important biological processes such as respiration, while chaperones and storage proteins capture intracellular copper cations. Copper transits from the extracellular compartment to the cytosol, through porins or other membrane transporters. Enzymes or chemical chelators, such as glutathione, also manipulate the redox state of copper to allow chaperone loading and extrusion in case of excess. Periplasmic and cytoplasmic chaperones provide copper for cuprotein biogenesis or deliver it to export systems. These export systems involve P1B-type ATPases or Resistance-Nodulation-Division (RND)-family multidrug efflux pumps. Periplasmic multi-copper oxidases also contribute significantly to periplasmic detoxification, by oxidizing Cu(I) to Cu(II) they reduce the metal's toxicity [2,5]. Synthesis of these molecular machines to ensure an equilibrium between cellular uptake and detoxification is mainly orchestrated by the actions of two types of regulators, MerR-like activators such as CueR that respond to cytoplasmic Cu(I), and two-component regulatory systems (TCSs) that sense periplasmic Cu(I), like the CusRS system [1,2,5]. TCSs are widespread in bacteria and play key roles in monitoring and triggering an adaptive response to environmental cues [6–8]. They comprise sensory histidine kinase (HK) proteins and cognate response regulators (RRs) communicating by phosphotransfer. Phosphorylation of RRs often results in modulation of gene expression, as most RRs are transcription factors. The copper resistance of bacterial species is also highly dependent on their genetic diversity. Indeed, additional determinants can be acquired through horizontal gene transfer (HGT), and these acquisitions tend to increase bacterial survival under copper stress [9,10]. For example, the 13-gene 37-kb GI-7 islet found in

some *Pseudomonas aeruginosa* isolates was found to provide them a copper-resistant phenotype that plays a key role in bacterial colonization and persistence in hospital environments [10,11]. The accessory genes acquired are often associated with metal and/or antibiotic resistance, such as the *pco/sil* genes carried on the widely distributed Tn7-like structures or on the CHASRI (Copper Homeostasis and Silver Resistance Island) [9,10,12–15].

The focus of our study, *P. aeruginosa*, is a metabolically versatile Gram-negative bacterium that exhibits high resistance to copper [5]. This opportunistic animal and plant pathogen is ubiquitously present in the environment, in soil and aquatic habitats. It is one of the major causes of nosocomial infections, establishing acute and chronic infections in immunocompromized patients [16,17]. Importantly, the ability of this bacterium to cope with copper stress was shown to be essential for its pathogenicity in a mouse model [18]. Like other bacteria, *P. aeruginosa* uses several strategies and systems to resist copper poisoning and cope with varying concentrations encountered in its environment [5,19]. Expression of the copper-resistance genes is mainly orchestrated by the cytoplasmic CueR regulator and a TCS, CopRS [19,20]. The cytoplasmic Cu(I)-sensing CueR is the "first responder", triggered after a copper shock, mainly to control genes whose products are involved in cytoplasmic copper tolerance, notably including an RND pump, the cytoplasmic chaperones CopZ1 and CopZ2, and CopA1. Upon copper binding, the repressive effect of the DNA-bound apo-CueR is relieved and the copper-loaded regulator is able to induce target gene expression by altering DNA topology. When high periplasmic Cu(I)/Cu(II) levels persist, the "second responder" CopR activates genes involved in periplasmic detoxification, such as *pcoAB* and *ptrA* [5]. The physiological importance of CopRS and its targets in terms of bacterial persistence and virulence has been underlined in chronic and acute wound infections [21]. Recently, the TCS DbsRS was shown to be an additional contributor to the copper response and to copper resistance [22]. CopRS responds only to copper, whereas DbsRS also responds to silver, upregulating the expression of genes involved in protein disulfide bond formation, among others. Interestingly, DbsRS requires more than 4-fold higher levels of copper for activation compared to the other two regulatory TCSs [22]. A short pulse of high copper causes a general stress response in addition to specific copper responses [20]. This general response is probably due to the generation of reactive oxygen species. In contrast, when bacteria grow in a high-copper medium, passive transport functions are downregulated to reduce membrane permeability. Nevertheless, both acute and chronic high-copper conditions trigger the synthesis of targets such as the tripartite RND pump CzcCBA, copper-related proteins PcoAB and P-type ATPase CopA1, MexPQ-OpmE, and the three TCSs–CopRS, DbsRS, and CzcRS [20].

We identified an Integrative and Conjugative Element (ICE) in the genome of the urinary isolate IHMA87 [23], related to *Pseudomonas paraeruginosa* PA7 [24,25]. This ICE carries, among others, genes encoding a potential additional copper-responsive TCS, named CusRS, and predicted copper resistance proteins, whose regulation and role were investigated in this study. Our results indicated that this accessory TCS CusRS is deeply integrated into the core regulatory network for copper homeostasis at *i*) the signal level, by detecting and responding to copper, *ii*) the transcriptional level, as the expression of its genes is cross-regulated by several core regulators (CopR, CzcR) responding to distinct cues such as copper and zinc, and *iii*) the molecular level, as we found cross-talk to occur between non-cognate partners of the CopRS and CusRS TCSs. Cross-talk is observed *in vivo* in the absence of one partner HK, with the remaining HK controlling the activity of both RRs to some extent. Such cross-talk was not expected as it occurs between conserved and accessory TCSs, which also rely on different molecular mechanisms to control the phosphorylation state of the cognate RR, as the signal triggers either the phosphatase activity or autokinase activity of the HK. This important

discovery shows how regulatory genes that accompany physiologically relevant genes on mobile elements can play an important role in the conserved regulatory network of bacteria.

# Results

## CusRS is a copper-responsive and strain-specific TCS

The two divergently transcribed operons (*IHMA87_02165–66* and *IHMA87_02164–57*) present in the urinary isolate IHMA87 are reminiscent of the *copSR-copABGOFCDK* locus identified in the soil-based heavy metal-tolerant β-proteobacterium *Achromobacter* sp. AO22: the gene organization is indeed similar and the sequence identity of the predicted proteins ranges from 84.9 to 99.7% [26] (Fig 1A). These predicted operons are absent from reference strains of *P. aeruginosa* (PAO1, PA14) and from the taxonomic outlier PA7, the reference strain of a *P. aeruginosa* clade that has recently been proposed as a new species called *P. paraeruginosa* [25]. We have chosen the exolysin-secreting clinical isolate IHMA87 as a model several years ago because, unlike PA7, this strain is virulent and amenable to genetic manipulation [27]. The above-mentioned locus is present in a genomic island described hereafter. The regulatory genes *IHMA87_02165–66* encode a putative copper-responsive TCS that shares homology with *P. aeruginosa* CopRS. We named this TCS CusRS due to the higher identity of the predicted proteins with the *E. coli* CusR and CusS (70.7% and 41.5% amino acid identity, respectively) than with IHMA87 CopR and CopS (69% and 36.2% amino acid identity, respectively). For the *IHMA87_02164–57* genes, we used the "*pcoA2B2GOFCDK*" nomenclature, with "*A2 and B2*" as copies of *pcoA* and *pcoB* genes are already present in the core genome of *P. aeruginosa* and *P. paraeruginosa* (www.pseudomonas.com; [28]). The predicted operon codes for several proteins required for metal detoxification, including the multi-copper oxidase PcoA2, a putative P-type ATPase (PcoF) and copper-binding periplasmic proteins likely to play a role as chaperones (PcoC, PcoG and PcoK).

Previous DAPseq experiments showed that, *in vitro*, recombinant CusR mainly binds to the 254-bp intergenic sequence of the *cusR* and *pcoA2* genes of the IHMA87 genome [29]. The summit of the enriched DAPseq peak pointed to a perfect palindromic sequence "ATTCATn-nATGTAAT" found upstream of a putative *cusR* promoter identified by BPROM (www. softberry.com) (Fig 1B). This palindromic sequence has similarities to the CopR consensus sequence in *P. aeruginosa* (CTGACAn$_5$GTAAT) (PRODORIC, www.prodoric.de). CusR binding to the intergenic sequence was confirmed by Electrophoretic Mobility Shift Assays (EMSA), and its specificity was demonstrated by competition with an unlabeled probe (Fig 1C), suggesting that CusR might regulate its own expression as well as that of the genes transcribed in the opposite orientation.

To assess the copper-responsiveness of the two operons and the involvement of the CusRS system in this response, we analyzed the expression of transcriptional *lacZ* reporter fusions with 500-bp sequences upstream of the *cusR* and *pcoA2* coding sequences in the wild-type strain and the *cusR*-deleted mutant. For copper stress conditions, we exposed the bacteria to 0.5 mM CuSO$_4$ for 2.5 h. This concentration and duration do not affect bacterial viability [19]. Exposure of the wild-type strain to copper led to significant stimulation of expression of both *cusR* and *pcoA2* (Fig 1D,1E). In contrast, no induction of *pcoA2* was observed following exposure of the *cusR* mutant to copper (Fig 1E), and activation of the *cusR* promoter, while still detected, was reduced about two-fold (Fig 1D). CusR therefore appears to control expression of both operons in a copper-dependent manner, but *cusR* expression must also be controlled by another copper-responsive regulatory factor.

To assess the importance of promoter elements, we constructed and analyzed a set of *lacZ* fusions associated with *cusR* upstream sequences of various lengths. Deletions affecting the

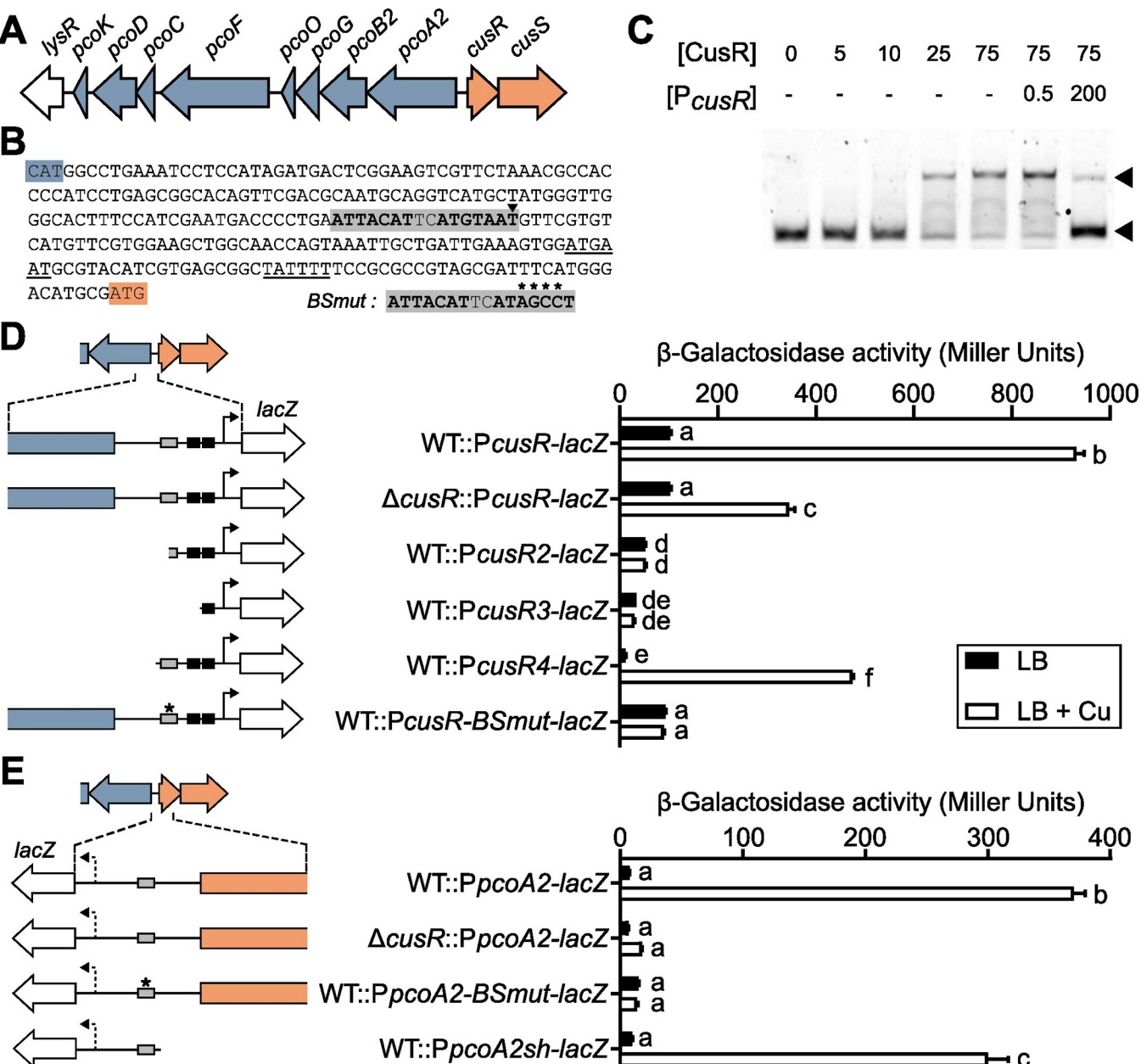

**Fig 1. CusRS is a copper responsive and self-regulated TCS.** (A) Genetic organization of the copper-related gene locus identified in strain IHMA87. Blue: copper-resistance genes; Orange: regulatory genes. (B) The *pcoA2-cusR* intergenic sequence, framed by the start codon (colored highlights) of each divergent gene. The -10/-35 sequences of the *cusR* promoter (underlined) were predicted using BPROM. The putative CusR binding site, the palindromic sequence, is in bold and highlighted in gray, with the summit of the peak identified by previous DAPseq [29] identified by an arrowhead. The mutated sequence (*BSmut*) used in the study is indicated in the inset, with the 4 mutated bases indicated by asterisks. (C) Electrophoretic mobility shift assay of CusR on the *pcoA2-cusR* intergenic region. Up to 75 nM of recombinant His6-CusR protein were incubated with 0.5 nM Cy5-labeled 60-mer probe for 15 min before electrophoresis. For competition assays, excess unlabeled P*cusR* probe (200 nM, 400-fold) was added to the reaction. Arrowheads indicate the positions of unbound free probes (bottom) and CusR/probe complexes (top). (D, E) β-galactosidase activities of the wild-type IHMA87 strain (WT) and the Δ*cusR* strain harboring the indicated P*cusR-lacZ* (D) and P*pcoA2- lacZ* (E) transcriptional fusions. The different regions studied are illustrated in the left panels. Black rectangles: putative -10/-35 boxes of *cusR* promoter; Gray rectangle: palindromic sequence, with the *BSmut* sequence indicated by an asterisk. Arrow: putative transcription start sites, the arrow is dotted for *pcoA2* as it was not predicted by BPROM. Experiments were performed in triplicate and error bars indicate the SEM. Different letters indicate significant differences according to two-way ANOVA followed by Tukey's multiple comparison test (*p*-value < 0.05).

palindromic sequence identified or the predicted -10/-35 boxes abolished expression, whereas a short fragment containing the palindrome maintained a copper response, but reduced both basal and activated expression levels (Fig 1D). Interestingly, mutation of a 4-bp sequence within the palindrome (ATTACATnnAT**AGCC**T) led to complete loss of the copper-responsiveness of the promoter. When examining the *pcoA2* promoter region (Fig 1E), shortening the sequence reduced the fold-change in expression following exposure to copper, whereas mutations in the palindrome led to the complete loss of copper-mediated P*pcoA2* activation. As expression of the *pcoA2* promoter is CusR-dependent, we concluded that CusR controls *pcoA2* expression by binding to the palindromic sequence identified.

To assess the extent of the *pco* operon, the mRNA levels of several genes within the locus (*pcoB2*, *pcoF*, and *pcoK*) and that of a downstream gene encoding a putative LysR-type transcription factor were measured by RTqPCR. Transcription was increased for all targets following copper stimulation, although the level of induction measured decreased with the distance between the gene and the *pcoA2* promoter (S1 Fig). In all cases, interruption of *pcoA2* transcription by introducing an omega interposon abolished the copper-inducing effect down to *lysR*, indicating that the last gene in the locus, *lysR*, is part of the same transcriptional unit.

Overall, these data show that the CusRS system in IHMA87 is functional and responds to copper. Following binding to a single site, CusR regulates its own expression and controls the expression of the nine-gene "*pcoA2*" operon, which contains genes that may play a role in copper resistance.

## CusR and the *pcoA2* operon contribute to copper resistance

Copper concentrations in urine are known to increase during UTIs, as part of an innate antimicrobial strategy [3,30]. Given the origin of IHMA87, we first questioned whether it had higher copper resistance compared to three reference strains. For these reference strains, we found *P. aeruginosa* PAO1 and PA14 to be more sensitive to 20 mM $CuSO_4$ than the *P. paraeruginosa* strain PA7 (S2 Fig). In these tests, IHMA87 had the highest resistance capacity, which could be provided by the additional, putative copper-resistance machinery, Pco, acquired through HGT. We also studied two others–JT87 and TA19, which like IHMA87 are members of the PA7-like group. These other strains showed variable resistance profiles, with TA19 being the most sensitive of all the strains tested. Thus, high copper resistance does not appear to be a common feature of UTI isolates.

Therefore, we investigated the role of the CusRS system and the *pcoA2* operon-encoded proteins in the high copper resistance of IHMA87. First, CusR was compared to the two well-described copper regulators, CopR and CueR. To do so, survival of individual, double, and triple mutants was analyzed on minimal medium in the presence of high copper concentrations (5 mM or 20 mM $CuSO_4$) and compared to that of the wild-type IHMA87 strain. As shown in Fig 2A, mutation of the *cueR* gene did not affect copper resistance whatever the genetic background, whereas the absence of CopR considerably reduced survival of the bacteria, revealing a key role of CopR and its regulon in the response to copper stress. When CopR was absent, the additional loss of CusR made the bacteria highly sensitive to copper toxicity. This sensitivity was observable at 5 mM $CuSO_4$ and accentuated at the highest concentration (Fig 2A), although the magnitude of the effect may vary somewhat between replicates, as shown in Fig 2B. To determine whether this phenotype was caused by the absence of the CusR regulator *per se* or by the lack of *pcoA2-lysR* expression, we introduced in several mutants an omega interposon into the *pcoA2* gene, stopping transcription of the operon from the first gene. This interposon strongly affected the copper resistance of the Δ*copR pcoA2*::Ω strain compared to that of the Δ*copR* mutant, with an effect observable from 5 mM $CuSO_4$ (Fig 2B). Thus, the proteins

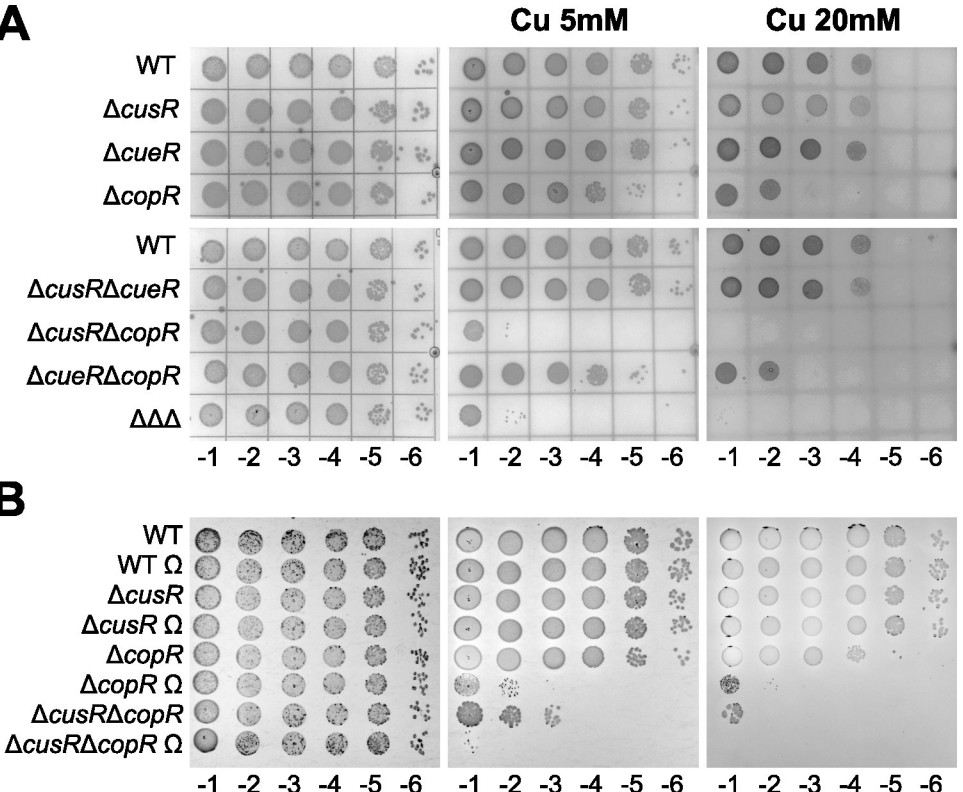

**Fig 2. The CusR regulon plays a role in bacterial copper resistance.** (A, B) Copper sensitivity plate assays of the different strains used in the study. ΔΔΔ corresponds to the Δ*cusR*Δ*cueR*Δ*copR* triple mutant and Ω corresponds to the *pcoA2*::Ω background. 10 µl of 10-fold serial dilutions, as indicated below the images ($10^{-1}$ to $10^{-6}$), were deposited on M9 plates containing the $CuSO_4$ concentration indicated. Plates were observed after 24 h (M9) or 48 h (Cu) of growth at 37˚C.

encoded by the CusR-regulated *pcoA2* operon are involved in copper detoxification and resistance. We can also see that *pcoA2*::Ω further increases the copper sensitivity of Δ*cusR*Δ*copR*, suggesting that the basal level of *pcoA2* expression in minimal medium and on plate is probably sufficient to confer some degree of copper resistance.

We further investigated the contribution to copper resistance of the different copper regulatory systems and the proteins encoded by the *pcoA2* operon. To do so, we followed the growth of appropriate mutant strains in lysogeny broth (LB) and compared it to IHMA87 wild-type growth. Firstly, increasing the copper concentration from 1 to 4 mM led to an extended lag phase for wild-type growth (S3A Fig). Significant cell death has been reported in the bacterial population after exposure to this high copper concentration for 2 h [19]. As the lag phase is a dynamic and adaptive period preparing bacteria for cell division, its extension indicated that more time is required both to repair altered cells and for physiological recuperation by stressed cells before exponential growth can occur [31,32]. Secondly, we observed that *cueR* and *copR* mutant growth was even more dramatically affected than wild-type growth in the presence of 4 mM $CuSO_4$. For these strains, especially Δ*copR*, the lag phase was considerably extended (S3A Fig). This increased delay before division indicated that the copper-resistance mechanisms controlled by the two regulators are important for survival in and adaptation to the stress. Interestingly, growth of Δ*cusR* and *pcoA2*::Ω mutants was unaffected, and was similar to IHMA87 growth, indicating that these mutations alone had no effect on the copper resistance. In contrast, the lack of both CopR and Pco proteins completely abolished bacterial growth

(S2A Fig). This result further underlined the important role in bacterial copper resistance played by the proteins encoded by the *pcoA2* operon. In the conditions used here, this role seems to be masked by proteins from the CopR regulon. Finally, we introduced an arabinose-inducible promoter (P*BAD*) into the chromosome, upstream of the *pcoA2* operon, and compared bacterial growth of the wild-type and modified strains over a range of copper concentrations. Increased expression of the operon conferred a clear growth advantage relative to wild-type bacteria, allowing faster adaptation to exposure to 4 mM $CuSO_4$, as indicated by the shorter lag phase (S3B Fig).

In conclusion, IHMA87 has a high intrinsic capacity to resist copper stress that can be further increased by overexpressing the *pcoA2* operon.

## Three TCSs cross-regulate the *cusR-pcoA2* intergenic region

As shown in Fig 1D, copper activates *cusR* expression even in the absence of CusR. As CopR binds *in vitro* to the *pcoA2-cusR* intergenic region [29], we assessed its role as well as that of the copper-responsive regulator CueR in the residual activity of P*cusR* and that of the divergent promoter. To do so, we used transcriptional *lacZ* fusions in several genetic backgrounds lacking one, two, or all three regulators. We found that CueR was dispensable in all the conditions tested (Fig 3A and 3B). However, induction of *cusRS* expression was reduced in the absence of CopR compared to the wild-type background, and was lost in the Δ*cusR*Δ*copR* mutant. This result confirmed CopR as the additional element responsible for the complete copper induction of *cusRS* in combination with CusR. Activation of the *cusRS* promoter by both CusR and CopR was abrogated by mutating the palindrome (Fig 1D), strongly supporting that the same intergenic binding site is required *in vivo* for activation by both regulators.

For the expression of the *pcoA2* operon, as mentioned above, no copper induction was observed in the Δ*cusR* strain (Figs 1E and 3B). Interestingly, *pcoA2* expression was higher in Δ*copR* than in the parental strain upon exposure to copper, suggesting a negative regulatory role of CopR. In contrast, expression of *pcoA2* genes was abolished in the Δ*cusR*Δ*copR* strain, indicating that the increased activation observed in the presence of copper in Δ*copR* was the result of CusR activity (Fig 3B). This observation suggested that CopR binding does not lead to transcriptional activation of the *pcoA2* promoter and, by competing for binding on the same palindrome, CopR seems to limit access for CusR, therefore reducing *pcoA2* activation.

To further explore cross-regulation between the different pathways, we tested whether CusR affects the expression of *copA1* and *pcoA*, which are respectively regulated by CueR and CopR [19]. Activation of the *copA1* promoter was abolished in the absence of CueR, but was unaffected by deletion of either *cusR* or *copR* (Fig 3C) as expected based on *in vitro* binding data [29]. *pcoA* expression had almost the opposite profile, with no effect observed following *cueR* deletion, but a complete lack of copper-induced expression in the *copR* mutant and a slight reduction in the copper-responsiveness in the *cusR* mutant (Fig 3D). This latter result was unexpected as *in vitro* results indicated that CusR does not bind to this promoter [29]. Suspecting an indirect effect, we measured expression of P*pcoA-lacZ* and P*pcoA2-lacZ* fusions in the *pcoA2*::Ω strain. Levels of both transcriptional fusions were decreased in the presence of copper compared to levels in the wild-type strain, whereas the level of CueR-regulated *copA* expression was similar for the two strains (Fig 3E). Based on these observations, the proteins encoded by the *pcoA2* operon may be required for the CopRS and CusRS systems to fully develop their copper-dependent activity. The effect is probably due to altered copper compartmentalization in the bacteria. These data also revealed an additional indirect layer of regulation, since the primary target of CusR, the *pcoA2* operon, seems important for both TCSs to mount a copper response.

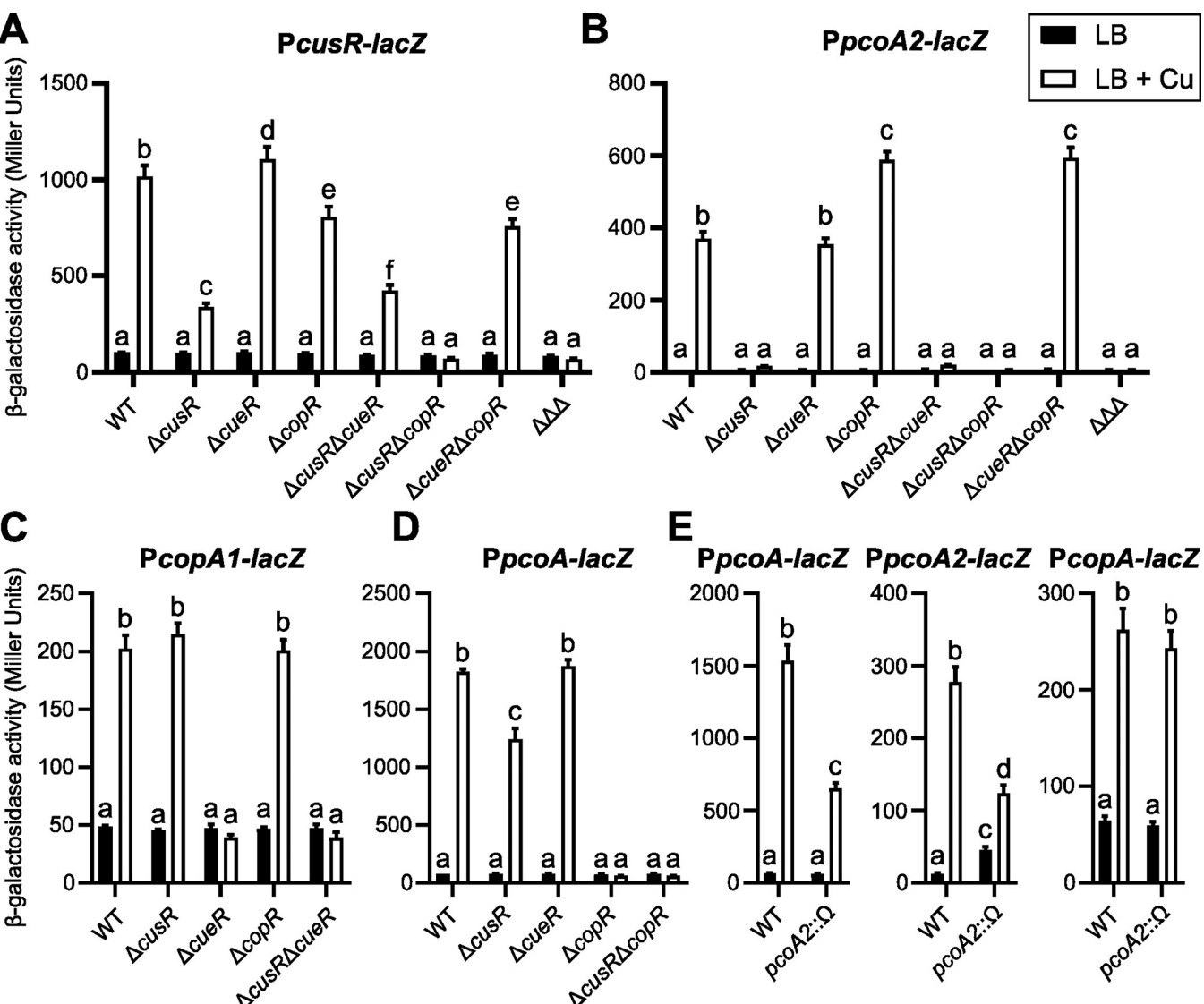

**Fig 3. Cross-regulation by CusRS and CopRS.** β-galactosidase activities are shown for strains harboring the P*cusR-lacZ* (A), P*pcoA2-lacZ* (B), P*copA1-lacZ* (C), or P*pcoA-lacZ* (D) transcriptional fusions. ΔΔΔ corresponds to the Δ*cusR*Δ*cueR*Δ*copR* triple mutant. (E) β-galactosidase activities of the wild-type and *pcoA2*::Ω strains containing the indicated transcriptional fusions. Experiments were performed in triplicate; error bars correspond to the SEM. Different letters indicate significant differences according to two-way ANOVA followed by Tukey's multiple comparison test ($p$-value < 0.05).

Mining and reanalyzing the DAPseq data [29] indicated that three other RRs strongly targeted the *pcoA2-cusR* intergenic sequence centered at the same position (S4A Fig). These RRs are the two poorly characterized regulators IrlR and MmnR, and the well-known zinc-responsive protein CzcR that binds to the sequence "GAAACn$_6$GTAAT" [33]. Additional target genes were found to be shared by CopR, CusR and CzcR (S4B, S4C Fig), presumably due to the similarity of their consensus sequences, especially the second half-site "GTAAT". Therefore, we analyzed the roles of these additional RRs on the expression of both the *cusRS* and *pcoA2* operons.

The TCS CzcRS is known to respond to several signals, including zinc stress [34,35]. We therefore assessed its effect on *pcoA2* and *cusRS* expression in the presence of copper and/or zinc. As a control of CzcR activity, we introduced a P*czcC-lacZ* fusion into the wild-type and

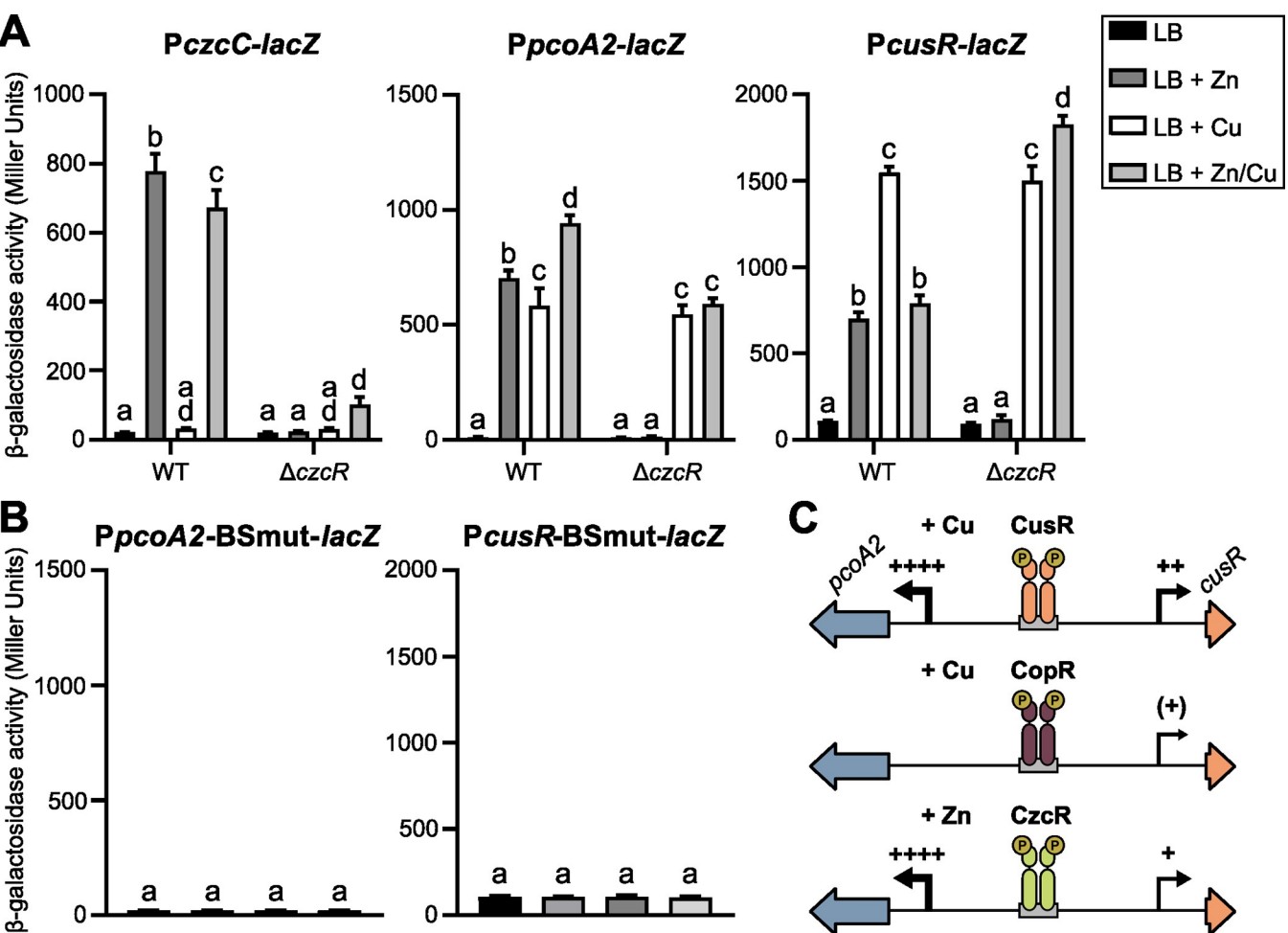

**Fig 4. CzcR regulates the *pcoA2-cusRS* intergenic region.** (A,B) β-galactosidase activities of the IHMA87 strain (WT) (A, B) and its isogenic *czcR* mutant (A) harboring the transcriptional fusions indicated. Activities were measured after 2.5 h of growth in LB or LB supplemented with 0.5 mM ZnCl₂, 0.5 mM CuSO₄, or 0.5 mM ZnCl₂/0.5 mM CuSO₄. Error bars indicate the SEM. Different letters indicate significant differences according to two-way ANOVA followed by Tukey's multiple comparison test (*p*-value < 0.05). (C) Model proposed for the action of CusR, CopR and CzcR, that compete for binding to the same site in the *pcoA2-cusR* intergenic region. The activating molecule (copper or zinc) is indicated for each system. The thickness of the long arrows and the number of « + » signs indicate the strength of transcriptional induction, if any.

Δ*czcR* strains, as a known CzcR target. Fig 4A shows that, as expected, the *czcC* promoter was upregulated after the addition of zinc, with or without the addition of copper, and that this upregulation was CzcR-dependent. Furthermore, the expression of *pcoA2* was activated by zinc to almost the same level as by copper. Exposure to both stimuli was slightly, but significantly, more efficient. Zinc activation was lost in the absence of CzcR, and the *pcoA2* expression level dropped to that observed in the wild-type in the presence of copper alone (Fig 4A). Therefore, both CusR and CzcR control and activate expression of the *pcoA2* operon to similar extents in response to copper and zinc, respectively. A distinct expression profile was observed for the *cusR* promoter: while copper strongly activated its expression, the induction following exposure to zinc was lower. In addition, zinc bypassed the copper effect when bacteria were simultaneously exposed to both metals. In the absence of CzcR, this "restraining" effect was lost, and the high-copper-inducible effect was restored, suggesting that CzcR must conceal the CusR binding site. To support the idea that CzcR binds to the same site as CusR, the expression of the two transcriptional fusions carrying the mutated binding site (Fig 1B, 1D and 1E) was

analyzed in the presence of copper and/or zinc in the wild-type strain. Not only was the expression of the two operons no longer activated by copper, as observed in Fig 4B, but the activation by zinc was also abolished. This strongly supports that CzcR directly controls the expression of the *cusRS* and *pcoA2* operons by binding to the palindromic site recognized by both CusR and CopR.

We then assessed a potential regulatory role of MmnR and IrlR and observed no *mmnR* or *irlR* deletion phenotypes in terms of *pcoA2* operon and *cusR* expression, even in the presence of copper (S5 Fig). However, we cannot exclude that this was due to an absence of signal and activation of the sensor in our experimental conditions.

Overall, the data indicate that *cusRS* and *pcoA2* operons are controlled *in vivo* by at least three TCSs: CusRS, CopRS, and CzcRS. The three RRs bind to the same palindromic site but modulate expression of the divergent operons differently. Indeed, while CusR and CzcR activate both operons, even if the latter is less effective on the *pcoA2* promoter, CopR only activates *cusRS*, repressing the *pcoA2* operon by hindering the positive action of CusR (Fig 4C).

## Molecular cross-talk between the CopRS and CusRS systems

It has been reported that the *P. aeruginosa* CopRS system functions in an unusual way [36]. Rather than relying on a conventional "phosphorylation-based mechanism" in which the sensor HK autophosphorylates upon detection of a signal, allowing subsequent phosphorylation of the cognate RR, the copper signal switches off the phosphatase activity of CopS towards its RR CopR. Thus, exposure to copper allows CopR to be phosphorylated and to activate its target genes. The mechanism leading to the phosphorylation of CopR remains unclear, as the phosphodonor has not yet been identified [36]. Our data, shown in Fig 5A, were consistent with the "phosphatase-based mechanism" of the CopRS system in the *P. paraeruginosa* strain IHMA87. Indeed, in a *cusRS* background, the absence of CopS led to a high-copper-blind expression of *pcoA* whereas expression of the sensor $CopS_{H235A}$ mutant protein with constitutive phosphatase activity [36] abolished *pcoA* expression. The copper-blind expression in $\Delta copS\Delta cusRS$ was approximately 5-fold higher than that observed in the copper-induced wild-type, suggesting that phosphatase activity of CopS is not completely switched off by the presence of copper in the wild-type background and that CopR is similarly efficiently phosphorylated in $\Delta copS\Delta cusRS$ regardless of the presence of metal. The same pattern was observed for *cusR* expression (Fig 5A). This result also confirmed that CopR can activate expression of this promoter in a $\Delta cusRS$ background, as previously suggested by the data presented in Fig 3A.

We then analyzed the functioning of the CusRS system in a genetic background lacking the CopR and CopS proteins to avoid any possible protein interference. As shown in Fig 5B, neither *pcoA2* nor *cusRS* operons were activated in the strain lacking the sensor protein CusS. This result indicates both that this HK is required for CusR activation by phosphorylation, and that no other protein can phosphorylate CusR. We also confirmed that the phosphorylated form of CusR is the active form, because no gene expression was observed when its conserved phosphorylation site ($Asp_{51}$) was mutated to an alanine (Fig 5B). In addition, mutation of the autophosphorylation site of CusS ($His_{266}$) abolished the expression of the two operons, indicating that it is required for the system to function properly (autophosphorylation prior to phosphotransfer). Altogether, these data show that the CusRS system is a canonical TCS, functioning like the *E. coli* CusRS system, with the detection of a signal triggering a phosphorylation cascade ultimately leading to target gene activation.

The two TCSs were then studied in different combinations of mutants of their HK gene to obtain information on possible cross-talk between non-cognate HKs and RRs. Focusing on a CopR-only-activated promoter, we found that deletion of *copS* resulted unexpectedly in an

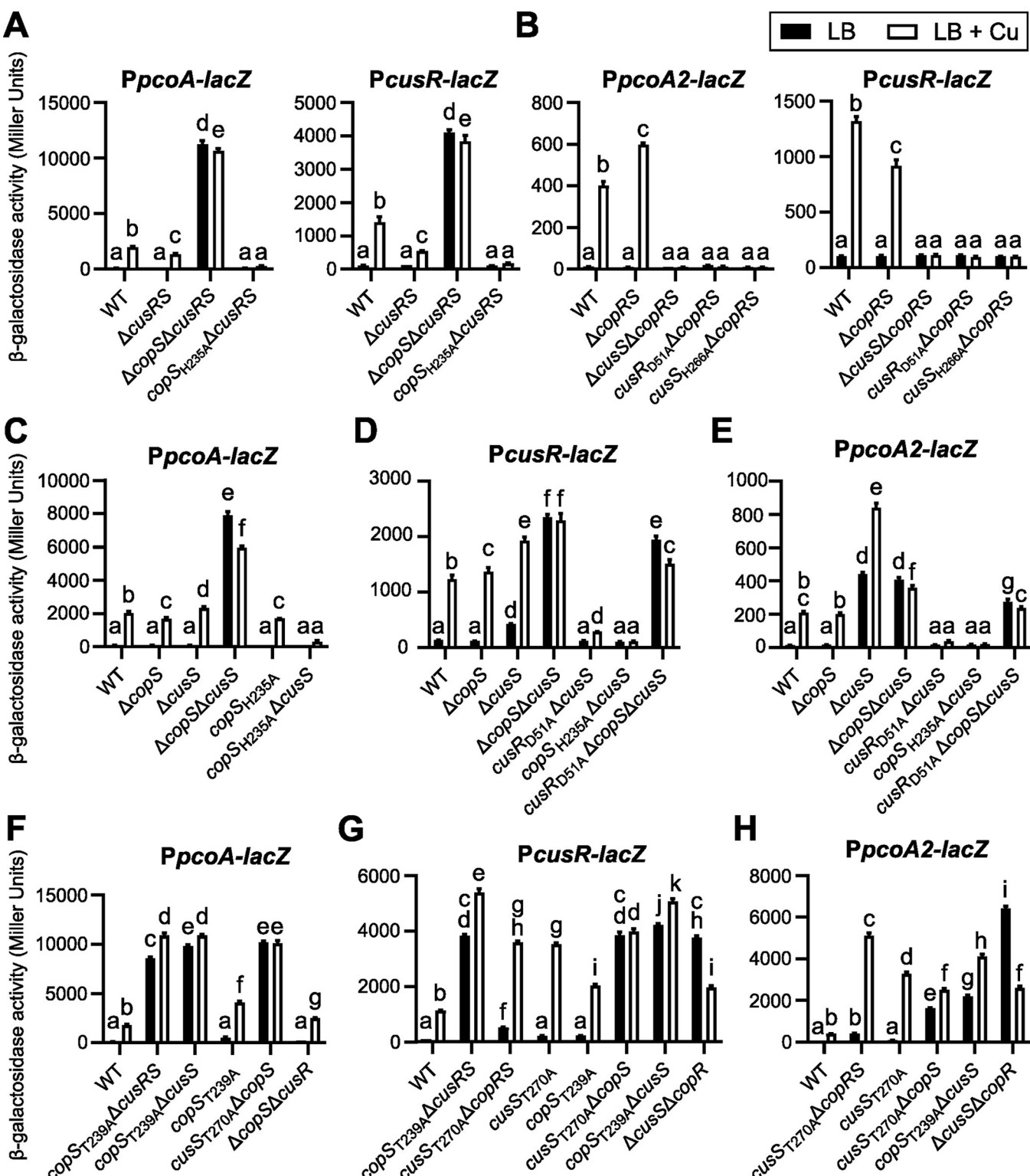

**Fig 5. *In vivo* cross-talk between CusRS and CopRS.** β-galactosidase activities of the strains harboring the transcriptional fusions indicated, after 2.5 h in LB or LB supplemented with 0.5 mM CuSO$_4$. Error bars indicate the SEM. Different letters indicate significant differences according to two-way ANOVA followed by Tukey's multiple comparison test (*p*-value < 0.05).

identical copper response pattern of P*pcoA-lacZ* expression to that observed in the wild-type strain or in a strain lacking the *cusS* gene (Fig 5C). However, deletion of both *copS* and *cusS* led to the strong constitutive activation of P*pcoA* (copper-blind expression) that was observed in the Δ*copS* mutant in the absence of CusRS (Fig 5A). These data suggest that the HK CusS certainly prevents *pcoA* expression by dephosphorylating CopR in the absence of both copper and CopS, suggesting that CusS can regulate CopR activity and compensate for the absence of the HK CopS. Alternatively, the interaction between CusS and CopR in the absence of CopS could prevent phosphorylation of CopR in the absence of signal. We further observed that the control of the phosphorylation state of CopR by CusS does not require the absence of its cognate HK. Indeed, normal expression of P*pcoA* was observed when CopS$_{H235A}$ was produced (Fig 5C), suggesting that CusS allows CopR phosphorylation upon copper sensing, bypassing the constitutive phosphatase activity of CopS$_{H235A}$.

To determine whether CopS could also interact with CusRS, we used P*cusR-lacZ* (Fig 5D) and P*pcoA2-lacZ* (Fig 5E) transcriptional fusions in several mutant genetic backgrounds. As expected, absence of CopS had little effect on the two promoters: indeed, CusS could control the phosphorylation status of CusR and of CopR, as observed above, and both these proteins activated transcription only in response to the copper signal. However, when *cusS* was deleted, in the absence or presence of copper, a slight increase in P*cusR* expression levels was observed (Fig 5D), and the effect on P*pcoA2* expression was even more pronounced (Fig 5E). Considering prior observations on Fig 5B, such effect must result from the presence of CopRS. To determine whether this transcriptional activation was due to the activity of phosphorylated CusR, we mutated its phosphorylatable residue (Asp51), as has been done in the CopR protein [36]. Indeed, the expression of *cusR* and *pcoA2* was abolished when CusR$_{D51A}$ was synthesized in the Δ*cusS* background mutant. Promoter of *cusR* is still slightly subject to copper activation by CopR as observed in Fig 5A. Thus, CopS appears to permit CusR phosphorylation in the presence of copper, but also in its absence, and to control its phosphorylation status, although not as tightly as the cognate HK CusS. P*cusR* and P*pcoA2* activation was not observed in the *copS*$_{H235A}$Δ*cusS* strain, suggesting that CopS$_{H235A}$ maintains CusR dephosphorylated or that it has lost its ability to phosphorylate the RR (Fig 5D). Copper-blind expression of both P*cusR* and P*pcoA2* was observed in the Δ*cusS*Δ*copS* mutant. As CusR cannot be phosphorylated in bacteria lacking both sensors (CusS and CopS), we reasoned that the expression pattern observed on P*cusR* (Fig 5D) can be attributed mainly to dysregulated CopR activity (Fig 5C). The reason for the activation of P*pcoA2* is more speculative, but our results indicate that, in such a background, *cusR* is highly expressed and, consequently, the RR protein is highly produced, which could lead to its binding to its targets despite its unphosphorylated state. Consistent with this is the copper-blind expression of the promoter measured in the strain *cusR*$_{D51A}$Δ*copS*Δ*cusS* (Fig 5E). Furthermore, P*pcoA2* expression levels were lower in Δ*cusS*Δ*copS* compared to in Δ*cusS*, which underscores the negative effect of phosphorylated CopR binding to the promoter (Fig 5E). In the *copS*$_{H235A}$Δ*cusS* background, the *pcoA2* promoter was not expressed, as no CusR is synthesized (Fig 5D).

The phosphatase activity of HKs towards their cognate RRs has been shown to require a conserved Thr/Asn residue close to the autophosphorylation site [37,38]. Therefore, to better understand the functioning of the two TCSs and the extent of their cross-talk, we mutated Thr$_{239}$ of CopS and Thr$_{270}$ of CusS to potentially abolish their phosphatase activity. As shows Fig 5F, the production of CopS$_{T239A}$ in the absence of CusRS, but also of CusS alone, resulted in a high and copper-blind expression of the *pcoA-lacZ* fusion. The same observation was made for the *cusR* expression in the background *copS*$_{T239A}$Δ*cusRS* (Fig 5G). These data support the model proposed by Novoa-Aponte *et al.* [36], in which CopS dephosphorylates CopR in the absence of copper. On the other hand, when CopS$_{T239A}$ was produced in the presence of

the second TCS CusRS, *pcoA* expression was almost equivalent to that observed in the wild-type strain, consistent with the proposed cross-talk model in which CusS can take over and compensate for the absence of CopS (Fig 5C). The $Thr_{270}$ mutation in CusS also appears to inactivate the phosphatase activity of this sensor protein towards CopR, as suggested by the mutant protein that can no longer dephosphorylate CopR in the absence of CopS ($cusS_{T270A}\Delta$-*copS*), resulting in a copper-blind phenotype. The final evidence that CusS is able to control the phosphorylation state of CopR *in vivo* in a copper-dependent manner was provided by the analysis of *pcoA* expression, which showed wild-type regulation in $\Delta copS\Delta cusR$ (Fig 5F). Differences in the expression levels of the *pcoA-lacZ* fusion were observed in the different mutants (Fig 5A, 5C and 5F), indicating that the extent of RR phosphorylation/dephosphorylation probably depends on the number of partners present and the protein-protein interactions involved.

We then analyzed the effect of the phosphatase-deficient CopS and CusS proteins on the expression of the *cusRS* (Fig 5G) and *pcoA2* (Fig 5H) operons which are cross-regulated by CusR and CopR. We first observed that $CusS_{T270A}$ retains its ability to auto-phosphorylate and transfer its phosphate to CusR, as observed with the high copper induction of P*cusR-lacZ* (Fig 5G) and P*pcoA2-lacZ* (Fig 5H) fusions. For P*pcoA2*-lacZ, Fig 5G also shows that the presence of either phosphatase-deficient protein ($CusS_{T270A}$ or $CopS_{T239A}$) did not affect the copper-responsiveness of *cusR* expression, and the additional absence of the second HK resulted in a copper-blind phenotype, as observed in $\Delta copS\Delta cusS$ (5D): this is consistent with CusS being able to dephosphorylate CopR in the absence of copper. Analysis of P*pcoA2-lacZ* expression also showed copper-blind expression in the presence of wild-type and phosphatase-deficient HKs ($cusS_{T270A}\Delta copS$ and $copS_{T239A}\Delta cusS$), as observed in $\Delta copS\Delta cusS$ (Fig 5E), albeit at a higher level of expression (Fig 5H), consistent with the transcriptional activity of overproduced, unphosphorylated CusR, as proposed above. Finally, the interaction between CopS and CusR was clearly observed *in vivo* in the absence of CusS and CopR, with a high expression of the P*cusR-lacZ* and P*pcoA2-lacZ* fusions in $\Delta cusS\Delta copR$, an expression that is halved in the presence of copper. Indeed, CusR can obtain its phosphate from the remaining HK, CopS, to be active (Fig 5G and 5H). Therefore, in addition to its phosphatase activity, CopS must have an autokinase activity, which appears to be modulated by copper availability. Interestingly, copper seems to partially decrease rather than increase this activity. Although non-intuitive, unorthodox regulation of the kinase activity may be related to the fact that it does not seem to be relevant for the function of the TCS CopRS and the expression of the CopR regulon.

Finally, we wondered whether the CopRS and CusRS systems respond to and modulate their regulon within the same range of copper concentrations. We compared their individual sensitivities to copper and their sensitivity relative to that of the cytoplasmic detector CueR. To do so, we used *lacZ* fusions to monitor the expression of a specific target of each transcriptional regulator over time in response to different copper levels. We found CopRS and CusRS to display a similar sensitivity and response to copper, inducing their target in the presence of 0.5 mM $CuSO_4$. In contrast, CueR activates its target at lower copper concentrations (from 0.01 mM) (S6 Fig).

In conclusion, although they respond to the same signal, the CopS and CusS HKs do not use the same molecular mechanism to control their cognate RR, the former using a phosphatase activity in the absence of the signal, whereas the latter is a phosphodonor in its presence. However, in the absence of one of the HKs, the other can compensate and effectively control the phosphorylation state and therefore the activity of both RRs, with a greater efficiency observed for CusS: the crosstalk thus allows an almost normal expression of the copper resistance proteins.

## Horizontal acquisition of the *cusRS* and *pcoA2* operons

As mentioned above, the reference *P. paraeruginosa* strain PA7 lacks the *cusRS* and *pcoA2* operons. Analysis of the genomic environment surrounding this copper-related locus showed that it is present on a genomic island that presents the features of an ICE (Fig 6A). ICEs are large mobile genetic elements that are crucial for the genetic plasticity of bacteria [39,40].

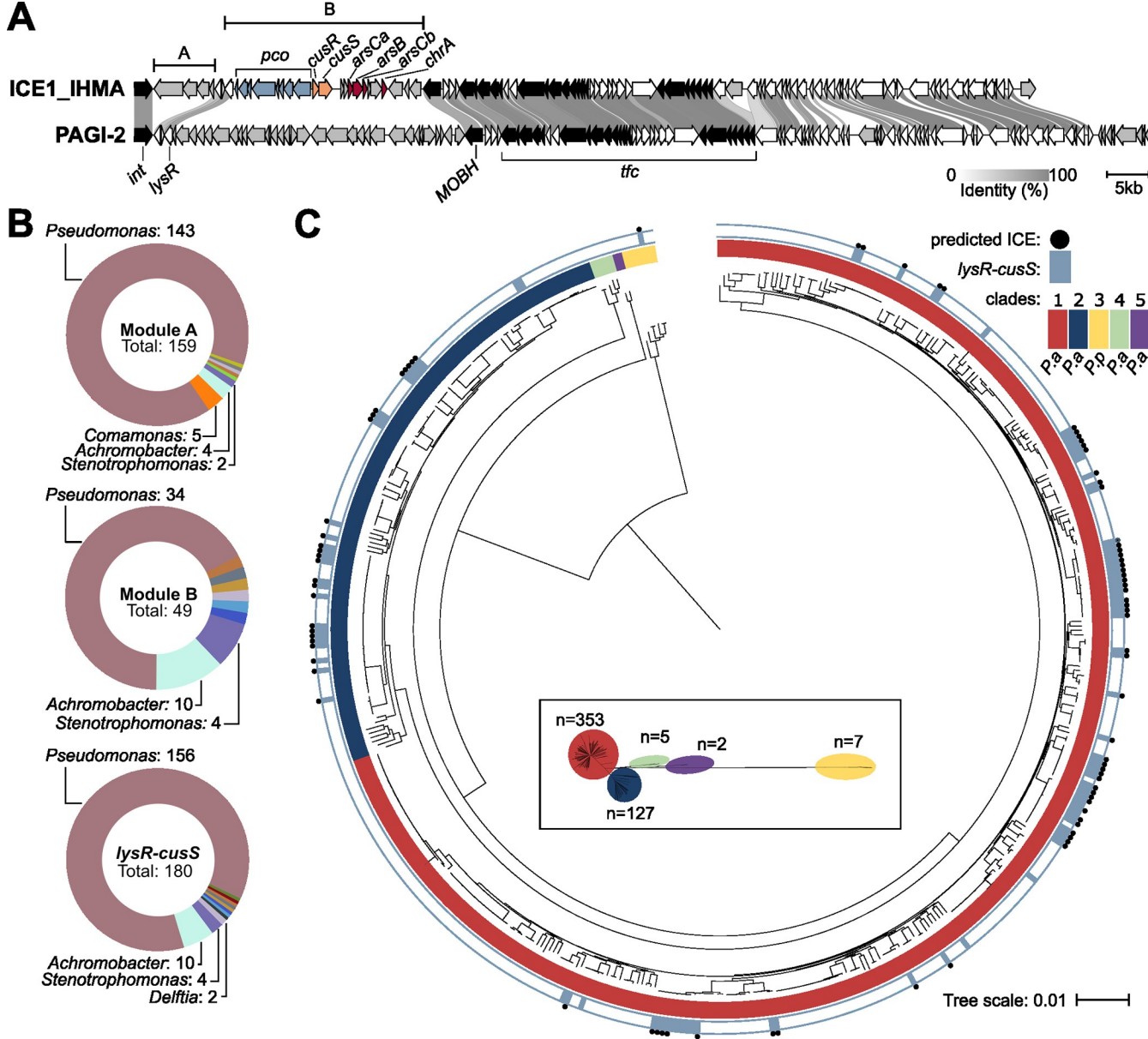

**Fig 6. Conservation of *lysR-cusS* locus in *P. aeruginosa* and *P. paraeruginosa* species.** (A) Comparison of the sequences of ICE1_IHMA and PAGI-2. ORFs are color-coded as follows: *pcoA2B2GOFCDK*, blue; *cusRS*, orange; genes involved in conjugation, transfer and integration of ICE, black; homologous genes, white; genes without any corresponding counterparts in ICE1_IHMA and PAGI-2, gray. Intensity of the link between ORFs was based on % identity between the orthologous proteins. (B) Distribution of genera across genomes harboring modules carried on ICE1_IHMA, as defined in (A). Genera with a single representative are not shown. Module A: *Delftia*, *Diaphorobacter*, *Ralstonia* and *Variovorax*; Module B: *Aerosticca*, *Delftia*, *Tistrella* and *Variovorax*; *lysR-cusS* locus: *Aerosticca*, *Bordetella*, *Delftia*, *Raoultella*, *Ralstonia*, *Tistrella*, *Variovorax* and *Xanthomonas*. (C) Phylogeny of *P. aeruginosa* and *P. paraeruginosa* species and distribution of *lysR-cusS*. Inset shows unrooted tree at scale, with the number of genomes for each clade. The presence of the locus is indicated by blue square, a blue dot signals whether the genes are encoded in a predicted ICE.

Consequently, the copper-related locus was probably acquired by HGT. We named this island ICE1_IHMA, and it is one of three predicted ICEs in the IHMA87 genome. It is 96,269 bp long, inserted at a tRNA-Gly site, and has an average GC content of 64%. Among the 100 predicted coding sequences present on ICE1_IHMA, some correspond to core ICE genes, required for the element's life cycle. These include signature genes coding for an integrase of phage origin (*int*) and a MOB(H)-type relaxase [41]. In addition, the element possesses a set of mating-pore formation (MPF) genes encoding a T4SS (*tfc* genes) including an ATPase (*tfc16/virB4*) and a Type IV coupling protein (*tfc6/t4cp2*). Among the eight types of MPF system–classed based on diverse features–ICE1_IHMA is a type G (MPF$_G$). These systems are frequently found in Proteobacteria [42]. ICEs also carry specific cargo genes, which usually provide the host with benefits and diverse phenotypic properties [39]. Based on sequence homology and the organization of its core genes, ICE1_IHMA is highly similar to PAGI-2, a prevalent *clc*-like family ICE in *P. aeruginosa* [16] (Fig 6A). By comparison with PAGI-2, we defined two specific cargo modules in ICE1_IHMA, named A and B. Module A comprises four genes, of which three (*IHMA87_02154–52*) are predicted to encode a CzcCBA-type heavy metal RND efflux pump. To assess whether this pump could be involved in copper efflux, we deleted *IHMA87_02154–52*. No effect was observed in terms of survival of the IHMA87Δ*RND* mutant obtained upon exposure to a high copper concentration (S2 Fig). In line with this result, the promoter region of *IHMA87_02154–52* was not targeted *in vitro* by CusR or CopR [29]. The other module, Module B, is composed of 21 cargo genes, among them the *cusRS* and *pcoA2* operons, but also several genes predicted to be involved in resistance to chromate and arsenic compounds. Copper, arsenic, and chromium resistance genes frequently co-occur in mobile genetic elements, favoring the spread of both metal resistance and antibiotic resistance genes. The spread of such resistance is of concern because co-selection promotes the acquisition of antibiotic resistance [10,43]. Although the gene encoding the LysR-type regulator is present on PAGI-2, it was considered to be part of Module B, since we demonstrated that it was part of the *pcoA2* operon.

As ICEs are modular and cargo genes can be exchanged, we assessed how the two cargo modules are distributed in bacterial genomes by performing blast searches against the NCBI database (Fig 6B). Modules A and B from ICE1_IHMA are both present in several bacteria from various genera. The most frequently represented genus was *Pseudomonas*, and the corresponding predominant species are *aeruginosa* and *paraeruginosa*. The two modules were also identified in other genera with distinct distributions, for example in a few *Achromobacter* and *Stenotrophomonas* genomes, whereas only Module A was identified in some *Comamonas* genomes. Interestingly, if the *lysR* gene located at the 3' extremity of the *pcoA2* operon is excluded from the search for Module B, identical results were obtained. This further emphasizes its co-occurrence with the *cusRS* and *pcoA2* operons, and suggests a possible role in their regulation.

We next focused our analysis on the distribution of the region from *lysR* to *cusS*, which comprises the entire sequence of the two operons (Fig 6B). Initially, this search provided more hits than searches using the whole of Module B, suggesting that this region can be acquired or maintained independently of the rest of the module. The genera harboring the locus were similar to those with the module B, but the distribution varied with a higher ratio of *Pseudomonas*. Among species, the region was mostly represented in *P. aeruginosa* and *P. paraeruginosa*, with only two strains of *Pseudomonas* not belonging to these species. We then examined the distribution and conservation of the *lysR-cusS* region over a database of 494 complete *P. aeruginosa* and *P. paraeruginosa* genomes. The clades were named based on the approach described by Freschi *et al.* [44] from 1 to 5, and, as proposed by Rudra *et al.* [25], outlier group 3 was (re) classified as *P. paraeruginosa* (*P.p*). Bioinformatics analysis revealed the locus to be present in

several strains of *P. aeruginosa* clades 1 and 2, and (as expected) in the IHMA87 strain, which is classed in clade 3 (Fig 6C). The locus was not detected in the few clade 4 and 5 strains for which genomes are available (www.pseudomonas.com; [28]).The locus was present as part of a putative ICE in over 80% of genomes. In the remaining 20% of genomes, it was part of regions bearing some similarity to an ICE, as determined based on neighboring genes and their predicted functions. The reason these regions were not immediately predicted to be ICEs can be attributed either to ICE degeneracy, or to a detection failure. In all the loci identified, the motif recognized by CusR in the *pcoA2-cusRS* intergenic sequence is conserved, indicating that the copper-resistance genes can be integrated into the bacterial regulatory network, allowing them to be readily expressed in their host under appropriate conditions.

Taken together, these data show that the *lysR-cusS* locus involved in copper resistance is mainly found in the *Pseudomonas* genus where it can be acquired by HGT as part of an ICE. As demonstrated by our research, the integration of this locus into the regulatory network involves cross-regulation and cross-talk with endogenous regulatory actors. These constraints must limit the acquisition of the locus by other genera.

## Discussion

In this study, we examined the integration of an acquired copper-resistance locus into a housekeeping system, both at the level of regulatory pathways and in terms of the protection it affords against toxic copper concentrations. The ICE-encoded copper-responsive TCS CusRS was shown to control expression of its own genes and a divergently transcribed nine-gene operon, *pcoA2B2GOFCDK-lysR*. The copper-related elements described here were initially acquired by HGT, and are not the only example of accessory copper genes found in *P. aeruginosa* species [11]. In bacteria, acquired systems can be even more efficient in coping with copper stress than the core mechanisms, as observed for the *copXL* locus in *Staphylococcus aureus* [45]. Although capable of conferring bacterial copper resistance, the results presented here showed that the positive effects of the *P. paraeruginosa* Pco proteins are masked by those of the core copper homeostasis proteins, controlled by the CopRS system [19]. However, mutation of the *pcoA2* operon reduced the metal response of both the CopRS and the CusRS regulatory systems (Fig 2B): this indicates that Pco proteins probably play a role in maintaining copper compartmentalization that affects signal detection by both acquired and conserved sensory HKs. Indeed, PcoA2 is a putative periplasmic multicopper oxidase oxidizing Cu(I), PcoB is a putative outer membrane exporter protein, PcoD is an inner membrane protein and PcoF is a putative P-type ATPase that extrudes copper from the cytosol. PcoC, PcoG and PcoK are predicted copper-binding periplasmic proteins that are likely to play a chaperone role, and PcoO may also be a chaperone but is located in the inner membrane. The Pco proteins can then be distributed in all compartments and membranes of the bacteria and participate in both periplasmic and cytosolic Cu detoxification, in particular by controlling copper shuttling. In addition, our results showed that the *pcoA2* operon contributes to overall copper homeostasis as it is under the control of the main core TCS CopRS. The control of accessory genes by housekeeping TCSs has previously been documented for copper homeostasis regulation. For example, in *E. coli*, although chromosome-encoded CusRS and plasmid-borne PcoRS systems are independent TCSs, CusR activates expression of the plasmid-encoded *pcoE* gene [46]. Similarly, in emerging pathogenic *S. aureus* strains, the core-encoded regulator CsoR controls expression of genes present in several mobile elements, conferring a phenotype that is hyper tolerant to copper [45,47]. In the *P. paraeruginosa* strain studied here, we found CopR to downregulate expression of the *pcoA2* operon instead of activating it, probably by competing with CusR for DNA binding (Fig 3B). This finding was unexpected as the *pcoA2* operon does

not appear to be a poisoned gift, given that it can improve the bacteria's resistance to copper stress when artificially overexpressed. We can hypothesize that the conserved copper resistance proteins controlled by the CopRS system are so efficient that an interaction between CopR and RNA polymerase has not been selected during evolution to activate the *pcoA2* operon since the acquisition of ICE. It is also possible that the fact that both CopRS and CusRS respond to the same signal (the same range of copper levels) did not prone CopR to activate expression of accessory genes already controlled by CusRS.

Although the homologous CopRS and CusRS systems respond to the same molecule over the same concentration range, their downstream signaling mechanisms differ. The HKs are often bifunctional enzymes with both phosphatase and autokinase activities, differentially triggered depending on the signal detection [48]. By analyzing a phosphatase-deficient CopS protein, we confirmed that the CopRS system relies on a "phosphatase mechanism" in *P. paraeruginosa* like that observed in *P. aeruginosa* [36], where CopS dephosphorylates CopR. The copper signal shuts down the phosphatase activity of CopS by a still unknown mechanism and, consequently, CopR becomes active, triggering gene expression (model in Fig 7). With this *modus operandi*, absence of HK leads to a copper-blind activation of the CopR regulon [36] that we also observed in a phosphatase-deficient CopS protein (Fig 5F). In the strain studied here, the CusRS system acted like a canonical TCS (Fig 7), functioning more like the recently discovered copper-responsive DbsRS TCS in *P. aeruginosa* [22]. DsbS can dephosphorylate its partner DsbR and, upon direct binding of copper, it can autophosphorylate. The phosphate is subsequently transferred to DsbR, which in turn activates target genes, notably those responsible for copper resistance [22]. Although no CusR dephosphorylation was observed, the use of different mutants, including one that produces the phosphatase-deficient CusS$_{T270A}$ protein (Fig 5), indicated that CusS HK can dephosphorylate the non-cognate CopR in the absence of CopS.

The *P. paraeruginosa* IHMA87 proteins CusR and CusS share 70.7% and 41.5% sequence identity, respectively, with their *E. coli* homologs [28]. In addition, both CusR regulators have a perfectly palindromic DNA-binding site and a very restrictive regulon [46]. Only one significant target sequence was identified for IHMA87 CusR by DAPseq [29]. This sequence is the palindrome "ATTCATnnATGTAAT", located in the intergenic sequence *pcoA2-cusR*. It is perfectly conserved in the equivalent locus in *Achromobacter* sp. AO22 [26]. *E. coli* CusR preferentially binds to the palindrome "AAAATGACAAnnTTGTCATTTT" [46], located between the two divergent operons *cusRS* and *cusC(F)BA*. Although no other chromosomal binding site could be identified [49], *E. coli* CusR also directly controls expression of the plasmid-borne gene *pcoE*, which has palindromic sequence (TGACAAnnTTGTCAT) in its promoter. Another feature of the *E. coli* CusRS system is its ability to confer resistance to silver toxicity. Thus, the HK CusS was shown to bind to both copper and silver [50,51]. As observed for chromosomally-encoded CusRS and plasmid-encoded PcoRS systems in *E. coli* [46], CopRS and CusRS are independent systems. However, in addition to the known cross-regulation (at the level of both gene target and signal) [48], we demonstrated *in vivo* molecular communication between the unrelated HK-RR pairs (Fig 7). Variable levels of cross-talk within TCS signaling pathways have also been reported in other bacteria, ranging from absent (or not observed) to highly represented as in the case of *Mycobacterium tuberculosis* [52,53]. As TCSs often arose during evolution through gene duplication, the high levels of sequence identity shared by the paralogs may favor interactions. However, distinct mechanisms normally evolve rapidly after duplication to insulate the systems, preventing inappropriate interactions and favoring the specificity of the molecular interactions within each system. Although generally considered to be a disadvantage for the bacteria, as it may decrease the specific response and trigger inappropriate responses, cross-talk can also be beneficial. Indeed, it can provide some evolutionary

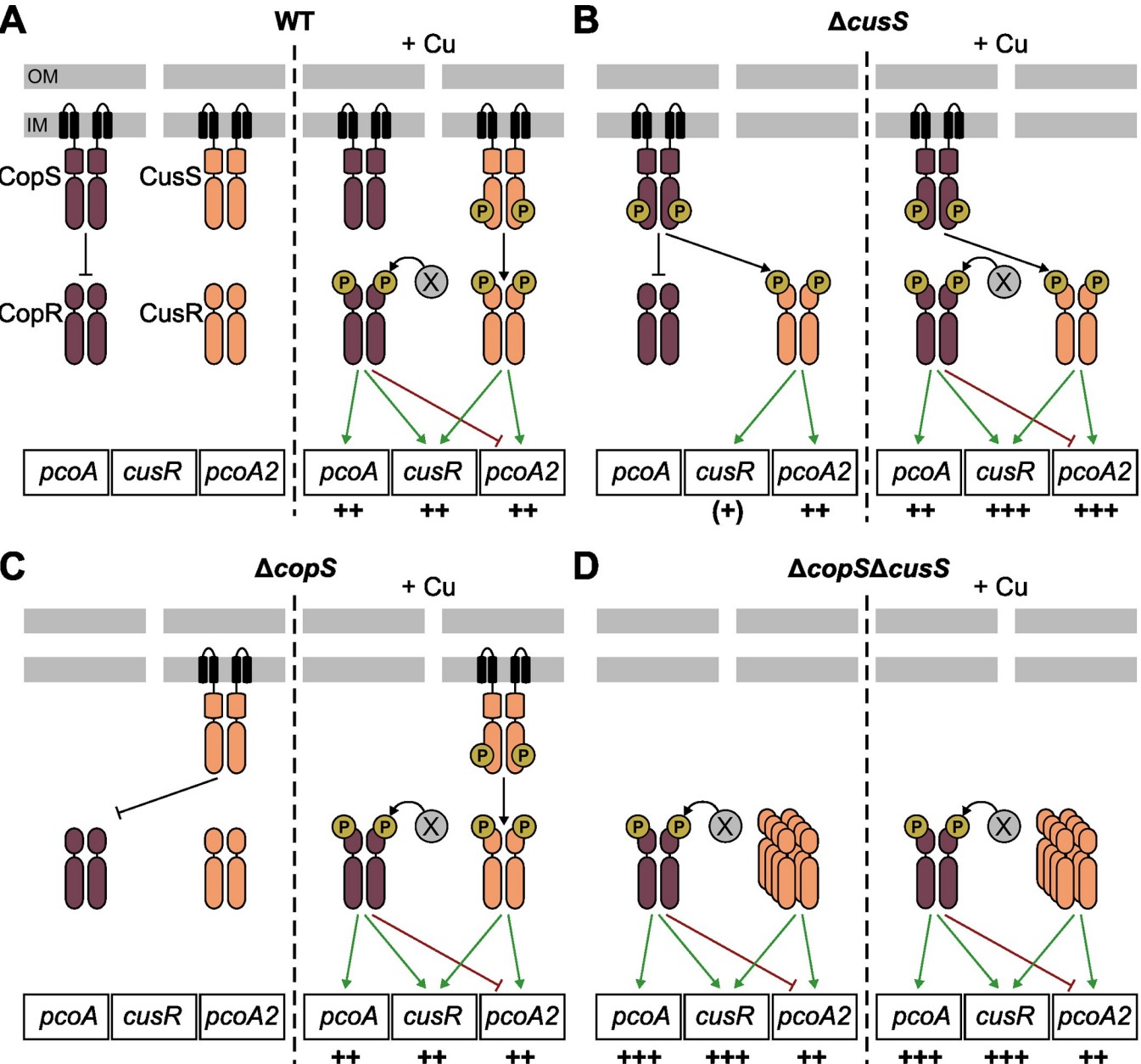

**Fig 7. Model of *in vivo* cross-talk between CopRS and CusRS.** Proposed mechanisms by which the two TCSs control the expression of the three target genes, *pcoA*, *cusR*, and *pcoA2* in the wild-type strain (A), Δ*cusS* (B), Δ*copS* (C), and Δ*copS*Δ*cusS* (D), in the absence or presence of copper. The HK either acts as a phosphatase (flat-headed black arrow) or as a phosphodonor (black arrow) towards the RR. X represents the unknown entity providing phosphate to CopR. Phosphorylated RRs generally activate the transcription of target genes (green arrow), with the number of "+" signs below the genes reflecting the extent of transcriptional activation. The flat-headed red arrow, observed with CopR on P*pcoA2*, represents the negative effect of the RR by preventing the binding of the CusR activator. Panel B: Although the phosphorylated state of CopS was unexpected, the data are consistent with a CopS-dependent phosphorylation of CusR in the absence of CusS.

benefits, such as complex signal-processing mechanisms or priming to respond to future stimuli [48,53,54]. Having four different TCSs for copper tolerance in probably allows the IHMA87 strain to fine-tune the expression of genes encoding the appropriate copper-related systems. Our results indicate that CusS could compensate for the absence of CopS by maintaining the relevant CopR phosphorylation state thanks to its copper-responsive function. This

first suggested that CusS is able to dephosphorylate CopR (Fig 5C), a hypothesis confirmed by using a phosphatase-deficient protein, CusS$_{T270A}$, which was no longer able to dephosphorylate the RR, resulting in a constant activation of the CopR-regulated gene (Fig 5F). Surprisingly, CusS could also control CopR activity in the presence of the constitutive phosphatase CopS$_{H235A}$ (Fig 5C). The reciprocal reaction was less efficient, as CopS was only partially capable of replacing CusS in its interaction with CusR. Whether the system also controls CusR phosphorylation is less clear, as this reaction requires a phosphotransfer step not reported for the CopS-CopR pair. Addressing this question with our genetic approach would require a kinase-deficient CopS protein which seems difficult to generate since mutation of the phosphorylation site of CopS results in a constitutive phosphatase protein, CopS$_{H235A}$ ([36], Fig 5A). In addition, results were blurred by cross-regulation exerted by the two TCSs on target genes (Fig 5D and 5E). We nevertheless produced schematic models that fitted the *in vivo* results indicating the cross-talk between CusRS and CopRS (Fig 7). These models will need to be consolidated by further experiments, in particular to determine the extent of the protein interactions and the HK activities toward the cognate/non-cognate RRs.

The *in vivo* cross-talk detected between the CusRS and CopRS systems in *P. paraeruginosa* was unexpected. First, CopRS is encoded by the core genome of the bacterium whereas CusRS is coded by genes present in the HGT-acquired ICE. Second, while the RRs of the two systems share 69% sequence identity, the HKs CopS and CusS only share 36.4% identity. These two HKs also act differently on their cognate RRs, with one removing the phosphate and the other providing it (Fig 7). Third, cross-talk usually involves the absence of both the reciprocal RR and the cognate HK of two distinct TCSs, as observed for CpxA–CpxR and EnvZ–OmpR in *E. coli* and mentioned for cross-talk from CusS to OmpR and CpxR [48,54]. The reason is that the RRs usually compete with other RRs for interaction with their cognate partner(s) to prevent misregulation. Here, however, in the absence of CopS, CusS is able to control the activity of CusR and CopR almost normally, as shown by the pattern of expression of their target genes in response to copper. Moreover, it did so even in the presence of the constitutive phosphatase protein CopS$_{H235A}$ or the phosphatase-deficient protein CopS$_{T239A}$. Although it originated from a lateral transfer, our bioinformatics analyses revealed that the locus is mainly present in *P. aeruginosa* and *P. paraeruginosa* strains, where the genes are probably readily expressed. The expression advantage of copper resistance on bacterial fitness might explain this distribution. In addition, the cross-regulation and cross-talk with endogenous regulatory TCSs may be required for appropriate integration of the locus, making it possible to tightly control expression without creating a burden. This would naturally limit spread of this ICE to other species.

Despite having been acquired by HGT, we also observed that the *cusRS-pcoA2* locus was a site of fierce competition for regulation, completely integrated into the conserved TCS regulatory network. Indeed, four core genome-encoded RRs–IrlR, MmnR, CzcR and CopR–in addition to the accessory CusR can bind the *cusRS-pcoA2* intergenic region *in vitro* [29]. All these RRs belong to the OmpR family with winged-HTH effector domains. The corresponding genes are organized in a two-gene operon with downstream, overlapping genes encoding the cognate HK (www.pseudomonas.com; [28]). Phylogenetic analysis clustered them together, inferring that they emerged by RR-HK pair duplication [55]. A genome-wide binding assay revealed these closely related regulators to share similar DNA-binding motifs, and extensively overlapping gene targets (S4 Fig) [29]. The physiological relevance of IrlR and MmnR binding could not be assessed here, as the signals detected by their cognate HK are as yet unknown. Although the *mmnRS* locus was reported to be upregulated in response to copper shock [56], we observed no effect of *mmnR* inactivation on the expression of *cusRS* and *pcoA2* in the presence of copper (S5 Fig).

Even if the five RRs bind to similar targets *in vitro*, their physiological role is more delimited, as CopR, CusR and CzcR are known to have their own regulon with only a few shared

and cross-regulated targets. Similarly, in *Pseudomonas stutzeri*, comparison of the *in vitro* binding sites and regulons of CzcR and the paralogs CopR1/CopR2 revealed limited target sharing and cross-regulation [57]. In addition, their binding was shown to induce either activation or repression with response amplitudes varying depending on the promoter bound [57]. Along the same lines, while CusR, CopR and CzcR could bind to the same DNA region, we observed that their effects on the expression of *cusRS-pcoA2* ranged from activation (to various extents) to no activation at all (illustrated in Fig 4C). The molecular dynamics of the TCSs, signal perception by HKs, and phosphorylated RR levels are among other important *in vivo* parameters that will need to be investigated. Nevertheless, based on the results presented here, these variable outcomes may also be due to differences in RR-DNA and/or RR-RNA polymerase interaction, as demonstrated for the *E. coli* paralogs KdpE and OmpR [58]. Indeed, a new set of target genes for duplicated RRs can evolve through changes in the DNA-binding sites recognized and their capacity to interact with the RNA polymerase [59]. However, sometimes the second mechanism is not without consequences, as shown here, where CopR not only fails to activate *pcoA2* promoter expression, but actually inhibits it by competing with CusR for binding. Furthermore, as proposed for CopRS and CzcRS in *P. stutzeri* [57], we cannot exclude the possibility that crosstalk between these two homologous TCSs exists, adding a level of complexity to the regulatory networks. Overall, our findings emphasized that the effects of a regulator binding to a promoter cannot always be predicted and must be determined on a case-by-case basis.

This study also illustrated the connection between zinc and copper homeostasis, with cross-regulation observed with CzcR, CopR and CusR (Fig 4C). The two metals play a major role in host-pathogen interactions and, during infection, higher copper and zinc concentrations can be encountered by bacteria under the same physiological conditions. For example, in phagolysosomes [60], these two metal ions contribute to bacterial killing. Synergized responses to the two elements relying on cross-regulation have been described in *P. aeruginosa* [61,62]. For instance, high copper concentrations lead to both increased zinc and copper resistance, and overproduced CopR can activate expression of *czcRS*, which in turn activates CzcCAB synthesis [61]. The regulator CzcR also controls the expression of the CopR-regulated *ptrA-czcE-queF* operon [34,62]. In a previous study, we found that PtrA is a periplasmic protein involved in copper tolerance [63]. In addition, CzcE was recently found to be a novel element required for zinc resistance [34]. Therefore, this operon harbors genes involved in tolerance to both metals. The data presented here further demonstrated that CzcR activates expression of the *pcoA2* operon, which is integrated in the RR regulon. As CzcRS is known to respond to cadmium in addition to zinc [34], this broadens the range of signals capable of triggering expression of the *pcoA2* and *cusRS* operons.

In conclusion, some *P. aeruginosa* strains harbor an ICE carrying genes for copper resistance, as well as cognate regulatory elements that are fully integrated into the bacterial regulatory network. The synthesis of this accessory copper system is tightly controlled in the intergenic regulatory region, which is a hot spot for binding of several RRs. Genetic analyses showed that at least three RRs compete for DNA binding and their docking stimulates or silences gene expression. The reason for the negative effect of the regulator CopR is not known, but expression of the *pcoA2* operon can be modulated by signals other than copper. Therefore, the accessory copper resistance mechanisms might be physiologically relevant in (as yet unidentified) conditions encountered in the host or in the environment.

## Material and methods

### Bacterial strains and growth conditions

The *E. coli*, *P. aeruginosa* and *P. paraeruginosa* strains used in this study are described in S1 Table. Bacteria were grown aerobically in LB at 37°C with agitation. To assess the effect of

copper on gene expression, overnight cultures were diluted to an $OD_{600}$ of 0.1 in LB supplemented with the indicated $CuSO_4$ concentrations, and growth was continued until the $OD_{600}$ reached around 1.0 (2.5 h). Growth in 96-well plates at 37˚C with agitation was assessed by following the $OD_{595}$ of 200-μL cultures in LB with the $CuSO_4$ concentrations indicated. Plates were read by a microplate reader at the indicated times. For copper sensitivity plate assays, 15 g/L agar was added to M9 minimal medium (48 mM $Na_2HPO_4$, 22 mM $KH_2PO_4$, 9 mM NaCl, 19 mM $NH_4Cl$, 2 mM $MgSO_4$, 0.1 mM $CaCl_2$, 0.2% Glucose) supplemented with $CuSO_4$. When LB cultures reached $OD_{600}$ of 1.0, 10 μL of a 10-fold serial dilution of the cultures were spotted on the M9 media and the plates were incubated for 1 to 2 days at 37˚C. Antibiotics were added at the following concentrations (in μg/mL): 200 spectinomycin (Sm), 100 ampicillin (Ap), 50 gentamicin (Gm), 10 tetracycline (Tc), and 10 chloramphenicol (Cm) for *E. coli*; 75 Gm and 75 Tc in presence of 5 irgasan (Irg) in LB for *P. aeruginosa*.

## Plasmids and genetic manipulation

The plasmids used and the primers for PCR are listed in S1 and S2 Tables, respectively. All constructions were verified by sequencing.

To generate *P. paraeruginosa* deletion mutants, upstream (sF1/sR1) and downstream (sF2/sR2) flanking regions of the different genes were fused and cloned into an *Sma*I-cut pEXG2 plasmid by sequence- and ligation-independent cloning (SLIC) using the appropriate primer pairs [64].

For chromosomal insertion of the P*BAD* upstream *pcoA2* operon, a pEXG2 plasmid was created by SLIC with upstream and downstream sequence of the *pcoA2* promoter sequence, creating a *Spe*I site 52 bp upstream of the start codon. Then, the 1580-bp "*araC*-P*BAD*" fragment was excised from pSW196 using *Xba*I and cloned into the *Spe*I site created, to produce pEXG2-P*BAD*-*pcoA2*-Sp.

For mutagenesis by insertion of the Ω interposon, the MCS-contained *Bam*HI site was first deleted from pEXG2 (pEXG2ΔB). Then a *de novo Bam*HI site was created while using SLIC to fuse two fragments of *pcoA2* in *Sma*I-cut pEXG2ΔB. Finally, the Ω interposon from pHP45Ω was introduced into *Bam*HI-cut pEXG2ΔB-*IHMA_02164*.

To replace the His or Thr residues in CusS and CopS with Ala, we used SLIC to generate a fragment with the mutated codon and to create a *Sac*II site (for His to Ala mutation) or *Bss*HII site (for Thr to Ala mutation) to allow subsequent restriction analysis screening of PCR-amplified fragments.

To create the transcriptional *lacZ* fusions, the miniCTX-*lacZ* plasmid was first modified to prevent transcriptional interference by integrating the strong *rrnB* terminator sequence into its *Xho*I-*Kpn*I sites, as described in Trouillon *et al.* [65], to produce miniCTX-T*rrnB*-*lacZ*. Then, fragments comprising the ATG and around 500 bp upstream of each target analyzed were amplified using the appropriate pairs (sF/sR) for insertion into *Sma*I-cut miniCTX-T*rrnB*-*lacZ* by SLIC. To create mutations in the regulator binding sites, another pair of overlapping mutated primers (sR1/sF2) were used to create two overlapping fragments (sF/sR1 et sF2/sR), also inserted into *Sma*I-cut miniCTX-T*rrnB*-*lacZ* by SLIC.

The pEXG2- and miniCTX-T*rrnB*-*lacZ*-derived vectors were transferred into *P. paraeruginosa* IHMA87 strains by triparental mating using the helper plasmid pRK600. Allelic exchanges for mutagenesis were selected as previously described [66]. To create the mutant encoding the $CusR_{D51A}$ protein, the mutated sequence was introduced into the Δ*cusR* mutants to replace the deleted gene.

For *E. coli* overproduction of CusR protein with an N-terminal 6xHis-Tag, the *cusR* gene was first amplified by PCR using IHMA87 genomic DNA as template, and SLIC was used to insert the fragment into the pET15b plasmid cut with *Nde*I and *Bam*HI.

## β-galactosidase activity assay

β-galactosidase activity was assessed at least in triplicate as previously described [67,68], after 2.5 h of growth ($OD_{600}$ of ~1.0) in the indicated media. Whole cells (0.5 ml) of each bacterial culture were permeabilized by addition of 20 μl of 0.1% SDS and 20 μl of chloroform, followed by vortexing for 1 min. Next, 0.1 or 0.2 ml of cells were removed and made up to 1ml with Z buffer (0.1 M $Na_2HPO_4$/$NaH_2PO_4$, 10 mM KCl, 1 mM $MgSO_4$, 50 mM 2-mercaptoethanol, pH 7.0), then placed at 28°C. Reaction was initiated by addition of 0.2 ml of o-nitrophenyl-β-D-galactopyranoside at 4 mg/ml and stopped with 0.5 ml of 1 M $Na_2CO_3$. The $OD_{420}$ was read after sedimentation of cell debris by centrifugation (5 min at 18,000g) and the activities were expressed in Miller units [$(OD_{420}x1000)(t_{min}xVol_{ml}xOD_{600})$]. On the bar graphs, activities were expressed in Miller Units (MU) and error bars correspond to the standard error of the mean (SEM). Statistical significance was assessed using ANOVA test with Tukey's method. S3 and S4 Tables provide all the numerical data and statistics underlying the graphs, respectively.

## Reverse Transcription-quantitative Polymerase Chain Reaction (RTqPCR)

Extraction of total RNA, cDNA synthesis and qPCR were carried out as previously described [65] with the following modifications: the cDNA was synthesized using 3 μg of RNA and the SuperScript IV first-strand synthesis system (Invitrogen). For each strain and condition, mRNA expression was measured for three biological replicates. Data were analyzed with the CFX Manager software (Bio-Rad), and expression levels were normalized relative to *rpoD* reference Cq values using the Pfaffl method. The sequences of the primers used can be found in S2 Table. Statistical analyses were performed by *T*-test.

## Production and purification of CusR

$His_6CusR$ ($H_6CusR$) was produced in *E. coli* BL21 Star (DE3) strains harboring pET15b-CusR, grown in LB at 37°C. Expression was induced by adding 1 mM isopropyl β-D-1-thiogalacto-pyranoside when cultures had reached an $OD_{600}$ of 0.6. Cells were grown for a further 2 h at 37°C and 150 rpm, and then harvested by centrifugation. They were resuspended in 20 mL of IMAC buffer (25 mM Tris-HCl, 500 mM NaCl, pH 8) containing 10 mM Imidazole and supplemented with Protease Inhibitor Mixture (Complete, Roche Applied Science) for lysis at 4°C by sonication. After centrifugation at $200,000 \times g$ for 30 min at 4°C, the soluble fraction was loaded onto a 1-mL anion exchange column (HiTrap 1 mL HP, GE Healthcare) and purified on an Akta purifier system. The protein was eluted by applying a 30-mL gradient ranging from 10 to 200 mM Imidazole in IMAC buffer at a flow rate of 1 mL/min. Eluted protein was diluted 1:20 in Tris-NaCl buffer (50 mM Tris- HCl, 150 mM NaCl, pH 8) for conservation before use.

## Electrophoretic Mobility Shift Assays (EMSA)

Probes were generated by annealing complementary primer pairs, one of which was Cy5-labeled, to form the fluorescent probe. 0.5 nM of the resulting 60-bp DNA fragments were incubated for 15 min at 25°C in EMSA Buffer (20 mM HEPES, 80 mM KCl, 20 mM $MgCl_2$, 0.8 mM EDTA, glycerol 7.5%, 0.1 mM dithiothreitol, pH 7.9). For competition assays, unlabeled DNA probes were added to the reaction at either 0.5 nM (equal amount) or 200 nM (400-fold excess). The proteins were then added at the indicated concentrations in a final reaction volume of 20 μL and incubated for a further 20 min at 25°C. Samples were then loaded onto a native 5% Tris-Borate-EDTA (TBE) polyacrylamide gel and run at 100 V and 4°C in cold 0.5X TBE Buffer. Fluorescence imaging was performed using an ImageQuant 800 imager.

## Bioinformatics analyses

ICEs were identified and delineated using the web-based tool ICEfinder [69]. ICEs were typed using ConjScan [70] on the Pasteur Galaxy platform [71]. Sequences were aligned and visualized with clinker [72]. Blast analyses were performed with BLASTn [73]. Complete genome sequences of *P. aeruginosa* (n = 487) and *P. paraeruginosa* (n = 7) were downloaded from the Pseudomonas Genome Database [28]. Genomes were annotated with Prokka [74], and core genomes were determined with Roary [75] before alignment with MAFFT using the default parameters [76] on the Galaxy server [77]. The resulting alignment was used to build a maximum-likelihood phylogenetic tree with FASTTREE [78] on the Galaxy server. The phylogenetic tree was finally visualized and annotated using iTOL [79].

## Supporting information

**S1 Fig. Delineation of the *pcoA2* operon.** RT-qPCR analysis of relative *pcoB2*, *pcoF*, *pcoK*, and *lysR* expression in IHMA87 (WT) and IHMA87 *pcoA2*::Ω. Strains were grown for 2.5 h in LB or LB containing 0.5 mM CuSO$_4$, as indicated. The *rpoD* gene was used as a reference. Experiments were performed in triplicate, and error bars represent the SEM. Significant differences with WT in LB according to unpaired *t*-test are annotated, ** *p*-value<0.01.
(TIF)

**S2 Fig. Copper resistance of different UTI isolates.** Copper sensitivity plate assays with the strains indicated. 10 μl of 10-fold serial dilutions, as indicated under the images ($10^0$ to $10^{-6}$), of each strain were deposited on M9 plates and M9 plates supplemented with 20 mM CuSO$_4$.
(TIF)

**S3 Fig. Copper resistance in rich media.** (A) Growth of the different strains was monitored in LB supplemented with various CuSO$_4$ concentrations, as indicated. (B) The P*BAD* promoter was introduced upstream of the *pcoA$_2$* operon to place it under the control of the inducible promoter. The growth of the resulting strain was compared to that of the wild-type strain in LB supplemented with 2% arabinose (to induce P*BAD*) and 0, 2, or 4 mM CuSO$_4$. (A,B) Growth in 96-well plates was monitored for the times indicated.
(TIF)

**S4 Fig. CusR, CopR and CzcR targets.** Reanalysis of previously published DAPseq data obtained for the IHMA87 genome [29]. (A) Enrichment coverage tracks of DAP-seq against negative controls are shown for the five RRs with binding sites on the copper-related locus, as indicated. The position of the summit of each peak from the translational start of *cusR* is given. (B) Venn diagram showing a large range of targets that overlap *in vitro* for the three RRs on the IHMA87 genome. Each target corresponds to the upstream region of one or more transcriptional units. Boundaries set for the upstream region were -400 and +20 bp from the ATG. (C) Targets common to the 3 RRs on the IHMA87 genome with the fold-enrichment of each peak and its rank among other targets of the RR. The targeted intergenic region of the different transcriptional units is indicated by a black triangle. The annotations "*03821*" and "*04274–76*" correspond to "*IHMA87_03821*" and "*IHMA87_04274_76*" (www.pseudomonas.com; [28]).
(TIF)

**S5 Fig. Effect of lack of IrlR and MmnR on the copper-related locus.** β-galactosidase activities of the indicated strains harboring the P*pcoA2-lacZ* or P*cusR-lacZ* transcriptional fusions, after 2.5 h in LB or LB supplemented with 0.5 mM CuSO$_4$. Experiments were performed in triplicate; error bars correspond to SEM. Different letters indicate significant differences

according to two-way ANOVA followed by Tukey's multiple comparison test ($p$-value $< 0.05$). (TIF)

**S6 Fig. Similar copper sensitivity of CusRS and CopRS.** β-galactosidase activities of the wild-type IHMA87 harboring the P*copA1-lacZ*, P*pcoA-lacZ*, or P*pcoA2-lacZ* transcriptional fusions. Growth in 96-well plates was assessed in LB supplemented with $CuSO_4$ at the concentrations indicated.
(TIF)

**S1 Table. Bacterial strains and plasmids used in this study.**
(DOCX)

**S2 Table. Primers used in this study.**
(DOCX)

**S3 Table. Numerical data underlying graphs.**
(XLSX)

**S4 Table. Summary statistics.**
(XLSX)

## Acknowledgments

We thank Eric Faudry for valuable advice on protein purification and Emmanuel Thévenon for giving us access to the ImageQuant imager. *P. aeruginosa* IHMA87 was obtained from the International Health Management Association, USA.

## Author Contributions

**Conceptualization:** Sylvie Elsen, Victor Simon.

**Formal analysis:** Victor Simon.

**Funding acquisition:** Ina Attrée.

**Investigation:** Sylvie Elsen, Victor Simon.

**Methodology:** Sylvie Elsen.

**Project administration:** Sylvie Elsen.

**Supervision:** Sylvie Elsen.

**Validation:** Sylvie Elsen, Victor Simon.

**Writing – original draft:** Sylvie Elsen, Victor Simon, Ina Attrée.

**Writing – review & editing:** Sylvie Elsen, Victor Simon.

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
