## [Decision Letter · Decision Letter 0]

3 May 2024

Dear Dr Elsen,

Thank you very much for submitting your Research Article entitled 'Cross-regulation and cross-talk of conserved and accessory two-component regulatory systems orchestrate Pseudomonas copper resistance' to PLOS Genetics.

The manuscript was fully evaluated at the editorial level and by independent peer reviewers. The reviewers appreciated the attention to an important topic but identified some concerns that we ask you address in a revised manuscript.

We therefore ask you to modify the manuscript according to the review recommendations. Your revisions should address the specific points made by each reviewer.

Yours sincerely,

Danielle A. Garsin

Academic Editor

PLOS Genetics

Sean Crosson

Section Editor

PLOS Genetics

Both reviewers are positive about this revised version of the manuscript. However, Reviewer #1 just had a couple more minor comments that I would like to see addressed.

Reviewer's Responses to Questions

**Comments to the Authors:**

Reviewer #1: In the previous version of this manuscript, the authors reported extensive cross-talks between the two component systems CusRS and CopRS. In this revision, the authors used phosphatase-deficient mutants of CusS and CopS to demonstrate their phosphatase activities towards cognate and non-cognate response regulators. These analyses much strengthen the model and the authors have addressed all my previous questions. However, there remain two minor points:

1. Please put the pcoA2::Ω strain back to Figure 2B. This is an important control for the ΔcusR and ΔcusR pcoA2::Ω phenotypes, because pcoA2 is regulated by multiple transcription factors.

2. Lines 440-445 still do not sufficiently explain why, in the ΔcusS ΔcopR strain, PcusR-lacZ and PpcoA2-lacZ is reduced by half in response to copper. It does appear that CopS can phosphorylate CusR, as written in the text. But why would basal expression be higher than copper-induced expression?

Reviewer #2: The authors addressed the concerns raised in my review of the former version of their manuscript (PGENETICS-D-23-01065 ). The message of the paper is now clearly stated and the explanations are more clear.

**Have all data underlying the figures and results presented in the manuscript been provided?**

Reviewer #1: Yes

Reviewer #2: Yes

PLOS authors have the option to publish the peer review history of their article (what does this mean?). If published, this will include your full peer review and any attached files.

Reviewer #1: No

Reviewer #2: No

---

## [Editor Report · Decision Letter 1]

29 May 2024

Dear Dr Elsen,

We are pleased to inform you that your manuscript entitled "Cross-regulation and cross-talk of conserved and accessory two-component regulatory systems orchestrate Pseudomonas copper resistance" has been editorially accepted for publication in PLOS Genetics. Congratulations!

Yours sincerely,

Danielle A. Garsin

Academic Editor

PLOS Genetics

Sean Crosson

Section Editor

PLOS Genetics

Comments from the reviewers (if applicable):

**Data Deposition**

http://datadryad.org/submit?journalID=pgenetics&manu=PGENETICS-D-24-00418R1

**Press Queries**

---

## [Editor Report · Acceptance letter]

6 Jun 2024

PGENETICS-D-24-00418R1 

Cross-regulation and cross-talk of conserved and accessory two-component regulatory systems orchestrate <i>Pseudomonas<i> copper resistance 

Dear Dr Elsen, 

We are pleased to inform you that your manuscript entitled "Cross-regulation and cross-talk of conserved and accessory two-component regulatory systems orchestrate <i>Pseudomonas<i> copper resistance" has been formally accepted for publication in PLOS Genetics! Your manuscript is now with our production department and you will be notified of the publication date in due course.

With kind regards,

Lilla Horvath

PLOS Genetics

On behalf of:
